# Risk Factors for Social Exclusion and Stigma in a Group of Non-Hospitalised Patients with COVID-19

**DOI:** 10.3390/ijerph22121775

**Published:** 2025-11-24

**Authors:** Georgina Becerril-Serna, Rafael Franco-Cendejas, Francisco Paz-Rodríguez

**Affiliations:** 1Outpatient Parenteral Drug Therapy, National Institute of Rehabilitation “Luis Guillermo Ibarra Ibarra”, Mexico City 14389, Mexico; georgina.becerrilse@anahuac.mx; 2Department of Agreement Preparation and Supervision, Institute of Security and Social Services for State Workers, Mexico City 06350, Mexico; 3Biomedical Research Deputy Directorate, National Institute of Rehabilitation “Luis Guillermo Ibarra Ibarra”, Mexico City 14389, Mexico; 4Clinical Neuropsychology Laboratory, National Institute of Neurology and Neurosurgery “Manuel Velasco Suárez”, Mexico City 14269, Mexico

**Keywords:** social stigma, social discrimination, COVID-19, SARS-CoV-2

## Abstract

Background: Stigma in many contexts is defined as a process of differentiation, othering, and discrimination. Its external manifestations include outward negative attitudes, and bad behaviour targeted at those with a specific condition. SARS-CoV-2 infections have been seen as a dangerous clinical condition towards people. The aim of the study was to describe the risk factors for suffering social exclusion and stigma in a group of people with COVID-19. Methods: We included 95 people into a descriptive, cross-sectional, and correlational study. They were invited to participate voluntarily to answer an online survey to determine and describe the effects produced by COVID-19 on social exclusion, stigma, self-compassion, demoralization, and self-care. Results: The participants, mostly women (61.1%), showed high education (72.6%) and did not require hospitalization (90.5%). A total of 18.9% of the participants reported social exclusion, of which 86.7% were women; they only had a dose of the COVID-19 vaccine (60%), poor sleep quality (46.3%), and suffered stigma (12.6%) from relatives and/or co-workers. Being female, not having the complete vaccination schedule and presenting stigma during COVID-19 were risk factors for social exclusion. Conclusion: Significant negative psychosocial consequences are observed in people who, in addition to suffering from COVID-19, suffer from social exclusion.

## 1. Introduction

Social determinants, such as healthcare access, income inequality, and cultural beliefs about medical tests or hospital procedures, may influence COVID-19 incidence and outcomes in vulnerable populations [1]. COVID-19 has had adverse effects on society and individuals. Among other things, stigma and its consequences for mental health have affected people’s quality of life. As a social construct, stigma is considered a daily experience of discrimination and rejection [2]. Furthermore, the impact can be long-lasting and continue even when symptoms no longer exist, or the person is no longer contagious to others. For this reason, it is said to be a personal experience or social process characterized by exclusion, rejection, guilt, and devaluation because of negative social judgments. Studies on previous outbreaks (SARS, Ebola) and COVID-19 show that a high proportion of survivors report stigmatization and the perception that they are still contagious, leading their family, friends and neighbours to avoid them [3,4]. The perception of threat to the population causes segregation and blame during COVID-19, which consists of rejection of coexistence, denial of services, or exclusion from work. This was directed at healthcare workers, people living on the streets or homeless, migrants, or people of Asian descent. According to the theory of terror management (the psychological need to assign a cause or enemy to the virus), it was based on fear of the unknown and contagion, looking for someone to blame. The emotional nature of fear reinforces stigmatizing beliefs. Fear of death (a central component of TMT) demonstrates the prevalence of mortality awareness, linking stigmatizing behaviour to actions aimed at avoiding the fear of death, resulting in greater prejudice and rejection [3]. Another study shows how the threat of SARS-CoV-2 quickly translates into xenophobia and prejudice directed toward an out-group perceived as the source [5]. Some research found that the fear caused by novelty and emergency was channelled through existing social prejudices (racism, xenophobia, ageism) and misinformation, resulting in discrimination and stigma [3,5]. Some researchers [5,6,7,8] have reported an increase in discriminatory behaviour and stigmatization at the beginning of the pandemic. Several theories explain the relationship between COVID-19 and stigma. Two focus on the fear of infection: one is related to the Behavioural Immune System (BIS), and the other to Terror Management Theory (TMT). BIS is a psychological mechanism that evolved to detect pathogenic signals in the environment (e.g., cough, fever, “unusual” appearances) and motivates avoidance behaviours as a precaution against infection. It is a psychological defence before the biological immune system can act. The person rejects and displays exaggerated social distancing toward any carrier and sometimes overgeneralizes to people who appear ill or belong to risk groups, even if they are not a real threat. In TMT, the awareness of one’s own mortality drives prejudice and the need for group belonging. During COVID-19, the high number of deaths reinforces identification with one’s own group and hostility toward what is considered or perceived as a threat, giving rise to xenophobia. In Labelling Theory (LT), stigma arises when a person or group is negatively labelled, leading to social exclusion and internalizing shame. The Stigma Linkage Model explains how stigma is linked to pre-existing social prejudices (race, class, geography, political orientation, etc.), as it does not occur in a vacuum but rather adheres to established social divisions and inequalities [9,10,11]. Thus, stigmatization during COVID-19 is a complex response driven by primal fear of infection and death (BIS, TMT), reinforced by social labels (LT), and directed toward groups already vulnerable due to pre-existing prejudices (SLM). Furthermore, the concepts of self-care and self-compassion also serve to understand coping strategies; according to Gilbert [12], compassion is associated with the biological capacity of care for others, sensitivity to discomfort, sympathy, tolerance to discomfort, empathy, non-judgement and sustaining a warm emotional tone. Self-compassion is the application of these capabilities in one’s own experience. According to Gilbert and Procter [13], self-pitying people are genuinely aware of their own well-being and are sensitive and empathetic to the discomfort of others, being able to be tolerant of discomfort without falling into self-criticism and judgement, understanding the causes of this discomfort, and treating themselves with warmth. It can often affect people’s mental health and lead to avoidance and distancing, further aggravating the situation [6,8]. The consequences of COVID-19 social determinants are people’s isolation and psychological burden, physical and mental violence, harassment, hiding the disease, and reduced care-seeking behaviour. The objective of this study was to describe the risk factors for suffering social exclusion and stigma in a group of people with COVID-19.

## 2. Materials and Methods

### 2.1. Procedures

This was a cross-sectional and descriptive study. Patients attending the Instituto Nacional de Rehabilitación “Luis Guillermo Ibarra Ibarra” in Mexico City were invited, agreed by consenting their participation to answer a questionnaire. The included patients answered a battery of scales designed ad hoc to assess the experience of social exclusion during their COVID-19 period, which was answered electronically while using a tablet or cell phone. If the participant needed help, an interviewer trained in this type of research provided support in the application of the battery of tests.

### 2.2. Sample

Between October 2021 and August 2022, a total of 95 convenience surveys were conducted among patients who had the infection but did not require invasive mechanical ventilation. The study included patients over 18 years of age who were discharged from hospital with a diagnosis of COVID-19 confirmed by a PCR test (oropharyngeal swab), following admission for moderate SARS-CoV-2 infection according to WHO criteria (moderate disease: presence of clinical signs of pneumonia [fever, cough, dyspnoea and/or tachycardia], with no signs of severe pneumonia and, in particular, oxygen saturation in ambient air [SaO_2_] ≥ 90%).

### 2.3. Battery of Tests

The battery of self-applicable tests was performed using Google Forms platform. It was divided into six sections: (1) sociodemographic and contextual information of the COVID-19 infection; (2) social exclusion; (3) stigma; (4) psychosocial affectations (self-compassion, demoralization); (5) Type D personality; and (6) sleep. Social exclusion was measured using a subjective relational approach, focusing on the breakdown of social ties (isolation, loneliness), discrimination, and rejection perceived by the individual. In this study, the following questions were asked: “Did you experience any discrimination while you had COVID-19?” and “What happened? What was the situation? Can you describe it?” Those who answered positively and met the subjective relational approach criteria were categorized as experiencing social exclusion. Perceived stigma was assessed using an adapted stigma scale [5,6], which measures two dimensions: internal stigma (distancing, exclusion, and shame caused by having the disease) and external stigma (degree of discomfort generated in social interactions). It consists of 20 items, each scored on a 4-point Likert scale. Higher scores indicate greater stigmatization. The scale has adequate reliability (Cronbach’s alpha = 0.96) and explains 57.5% of the variance. The Self-Compassion Scale (SCS) (26 items) was applied, assessing how the individual perceives their actions in difficult times. It is composed of six domains, in three distinct but theoretically related concepts: common humanity, mindfulness, and self-compassion (Self-Compassion positive), which have their counterparts in self-judgement, isolation, and over-identification (Self-Compassion negative) [14]. An overall SCS rating has adequate reliability and validity in Mexican population internal consistency (Cronbach alpha = 0.84) [15,16], even in different cultures [17]. The DS Demoralization Scale (24 items) was also used to explore the inability or incompetence to effectively cope with a stressful situation, through five domains: loss of meaning in life, dysphoria, discouragement, helplessness, and sense of failure [18]. Most studies [19] have found adequate levels of internal consistency (Cronbach alpha = 0.94). Type D personality has been described as the trend to experience an increased occurrence of negative affectivity and social inhibition and may negatively affect health status [20]. Poor sleep may cause disadvantageous decision-making and poor social functioning. Pittsburgh Sleep Quality Index (ICSP) assesses the quality of sleep through a self-applicable questionnaire of 24 questions, 19 of which are used to obtain an overall rating, which are rated on a scale of 0 to 3. A score >5 distinguishes those who sleep poorly from those who sleep well (Cronbach alpha = 0.78) [21].

### 2.4. Statistical Analysis

A descriptive analysis of the variables of interest was performed. A normal distribution was observed using the Kolmogorov–Smirnov test. The stigma scale was validated with FACTOR 11.5 software [22], and its compositional structure was explored using exploratory factor analysis (EFA). Since it is a Likert-type questionnaire, the ordinary least squares method was chosen [22]. The average partial least squares (ALS) method was used to determine the number of dimensions; for factor estimation, the unweighted least squares method was employed. A univariate analysis with nonparametric statistics (Wilcoxon and Kruskal–Wallis tests) explored the significant differences between the sample characteristics and the level of stigma during the COVID-19 pandemic. Finally, to identify risk factors, a contingency table analysis was performed, and crude prevalence ratios were estimated for categorical and ordinal variables in a stratified analysis, allowing for the identification of confounding variables and interactions between them. Subsequently, a multivariate analysis using unconditional logistic regression was conducted to estimate the probability of experiencing social exclusion based on the effect of covariates, using adjusted odds ratios. The model included all significant variables identified in the bivariate analysis. The most parsimonious model was selected using the Hosmer–Lemeshow goodness-of-fit test. Dummy variables were created to categorize ordinal variables. A *p*-value < 0.05 was considered statistically significant (two-tailed test). The data were analyzed using SPSS version 22.0.

### 2.5. Ethical Considerations

Participants signed a digital consent form before completing the questionnaires [23]. The institutional clinical research ethics committee approved the study with number 70/21 COVID.

## 3. Results

The study population’s general characteristics are shown in Table 1. Of the 95 participants in the study, 61.1% were women and 64.2% were between 30 and 50 years old, 72.6% had high education, 41% were single, and mostly did not require hospitalization (95.8%). The time they were with COVID-19 symptoms was an average of 2.5 (±1.8) weeks with a range of 1 to 10 weeks. The people who reported suffering social exclusion (discrimination) were women (83.3%); however, in the other variables, no significant differences were observed.

On the other hand, it was observed that those who reported having suffered social exclusion had a longer period with symptoms, since 61 participants (64.2%) presented symptoms for more than a week. Higher scores on the stigma scale were also present—both promulgated and perceived—differing from those who do not report social exclusion during the time they suffered COVID-19; even so, in 12 participants (12.6%), stigma could be observed. Regarding self-compassion and demoralization, no significant differences were observed, but higher scores were detected in those who perceived social exclusion during COVID-19 (Table 2).

In terms of stigma, it was observed that people who only had one dose of the vaccine had higher scores on the social stigma and the total stigma scale. In the same way, those who suffered social exclusion during the COVID-19 period showed differences in the scores of enacted and social stigmas, as well as in the total scale, obtaining higher scores. This same situation was observed in people who showed poor sleep quality assessed with the Pittsburgh sleep scale (Table 3).

Finally, the logistic regression model for social rejection (Table 4) showed that the variables giving risk of social exclusion during the pandemic were being a woman (*p* = 0.041), having a single vaccination dose (*p* = 0.035), and suffering stigma (*p* = 0.004). The Hosmer–Lemeshow test showed that the model presented a good fit (*p* = 0.641).

## 4. Discussion

As in other studies [3,4,13,24], people who suffered from COVID-19 perceived rejection when reintegrating into their daily activities [4]. We found in this study that 18.9% reported social exclusion, being mostly women, which agrees with the data reported in the EDIS 2021 [24], where it described that during the first year of the COVID-19 pandemic, an increase in discrimination against women was observed, and not being able to attend meetings was reported as the most frequent form of social exclusion [6]. Gender differentiation might be explained because women are exposed to more social disadvantages related to health access, so it is believed that females are more stigmatized than males for the same behaviour such as in other infectious diseases [25]. Therefore, such kinds of rejection might negatively influence internalized stigma.

In our study, people suffered rejection from colleagues, neighbours, or relatives; as an example, they commented that they did not want to take samples in the absence of serious symptoms or provide care, in addition to returning to work where they were avoided by their colleagues or bosses. Discrimination and social exclusion secondarily may have detrimental health consequences and could reduce the possibility of seeking help sooner because of social stigma [26]. On the other hand, and interestingly, they also had longer symptoms than people who did not report suffering social exclusion; this might be related to a delay in seeking clinical attention secondary to this suffering, perpetuating their symptoms, as it has been observed in other diseases such as HIV or psychiatric ones [27,28].

Our findings show greater internal and social stigma, which can be explained because they perceive themselves as highly vulnerable to the reactions of the environment where they live; they are afraid and uncertain, partly because several did not have the complete vaccination scheme (43.2%) before contracting COVID-19. As in other studies, people mentioned the infection became the cause of the problem of being stigmatized [10]. This also shows effects after suffering COVID-19, because 46.3% report poor sleep quality. Recent findings highlighted that individuals’ adaptability could affect sleep quality [29], as highly adaptive personalities had less perceived stress and better sleep quality compared to subjects with a lower degree of adaptability.

We must not forget that COVID-19 frequently occurs in low socioeconomic strata, among people with precarious or low-skilled jobs and living in poverty [30], which can contribute to clinical and social consequences. According to the study “COVID-19 Mortality in Mexico,” seven out of ten victims had only a primary education and lived in poverty [31]. In Pakistan, a high degree of stigma was observed [3] where the disease was considered the responsibility of the carrier. In Korea [4], post-traumatic stress disorder (PTSD), a history of psychiatric illness, and the stigma of COVID-19 infection, as well as the total duration of isolation, were identified as significant risk factors. Similarly, in Uganda, 8.4% suffered violence or discrimination during the COVID-19 pandemic [32]. The most frequent cause was their socioeconomic status. Our results are consistent with the results of previous studies and could be associated with the sociocultural context of a specific period, which could be modified due to the dynamic evolution of the pandemic.

In Mexico [5] in the general population, it was reported that during the first wave of COVID-19, 20.8% presented symptoms of anxiety and 27.5% of depression, according to the results of perception of the Survey on Discrimination in Mexico City (EDIS), prepared by the Council to Prevent and Eliminate Discrimination, COPRED [24]. The people surveyed consider that the most discriminated group are those with brown skin, followed by indigenous people and women being in third place, a group on which the perception of discrimination against them increased drastically, since it improved from 2.7% in 2013 to 4.3% in 2017 and now 9.4% in the EDIS 2021 [24]. One in four people surveyed (25%) have felt discriminated against at least once. Rossi et al. [33] reported that in hospitalized patients 26.7% presented anxiety and symptoms of demoralization, mainly related to sadness and anxiety with the current and future situation, perceived as confusing and uncertain. However, in our study, the statistical results did not show a significant association.

We believe patients are vulnerable to demoralization, especially if we consider structural factors that affect health (such as hopelessness), life chances, equality, and social justice, during and after COVID-19. Demoralization [34,35] is recognized as the inability or incompetence to effectively face a stressful situation, which causes discomfort and subjective incompetence. It is produced by the subjective perception of the person of damage to their independence and competence; in the case of COVID-19 this feeling is the result of uncertainty regarding the consequences of the disease, people’s expectations, and the loss of social roles, producing isolation, lack of control, and worry, and affecting self-efficacy and self-esteem. In patients with prolonged hospitalization and those who were isolated from family and friends, many psychological problems were observed requiring attention, with anxiety, demoralization, stress, depression, and mourning being the most frequent [29].

This study provides information on how to address gender disparities, poverty, and stigma to develop public health campaigns. Clinical interventions should aim to raise awareness among both healthcare workers and patients about the detection of stigma and social exclusion. Likewise, community education should be used to eliminate myths, misinformation, and prejudices to improve the psychosocial well-being of people affected by COVID-19. The findings of this study provide critical insights that can inform public health policies, clinical interventions, and community education initiatives to combat stigma and social exclusion among individuals affected by COVID-19

There are several limitations in the current study that must be mentioned. First, this study was a cross-sectional study, limiting the establishment of causal relationships between variables. Therefore, longitudinal studies are needed. Second, we measured COVID-19 related stigma in terms of negative experiences, emotional responses, sleep quality among only COVID-19 patients and not to the general population. Third, we conducted only quantitative analyses while using scales and not comprehensive assessments of stigma including qualitative analyses which could enrich the information. Fourth, as we work in a hospital, we included only patients who had access to medical services and not those who did not. Including the general population may allow a deeper understanding of COVID-19-related stigma. Fifth, the sample size of this study is a key factor to be considered in the interpretation of results that could have potentially introduced several biases.

## 5. Conclusions

In essence, the stigmatization during this period was based on fear of the unknown and of contagion, which translated into a search for infected people to blame and the amplification of pre-existing prejudices. Thus, the stigmatization during COVID-19 is a complex response driven by primal fear of infection and death (BIS, TMT), reinforced by social labels (LT), and directed toward groups already vulnerable due to pre-existing prejudices (SLM). Significant negative psychosocial consequences are observed in people who, in addition to suffering from COVID-19, suffer from social exclusion. Stigma could be associated with a longer symptomatic duration, and women might be more affected.

## Figures and Tables

**Table 1 ijerph-22-01775-t001:** Demographic characteristics compared by social exclusion.

	Social Exclusion			Total
	Yes = 18 *n* (%)	No = 77 *n* (%)	n = 95 *n* (%)	F (*p*)
Gender				4.636 (0.031)
Male	3 (16.7)	34 (44.2)	37 (38.9)	
Female	15 (83.3)	43 (55.8)	58 (61.1)	
Age				1.958 (0.376)
18–29	2 (11.1)	19 (24.7)	21 (22.1)	
30–50	14 (77.8)	47 (61.0)	61 (64.2)	
51+	2 (11.1)	11 (14.3)	13 (13.7)	
Education				1.228 (0.525)
High school	3 (16.7)	23 (29.9)	26 (27.4)	
University	11 (61.1)	39 (50.6)	50 (52.6)	
Postgraduate	4 (22.2)	15 (19.5)	19 (20.0)	
Marital status				6.190 (0.103)
Single	3 (16.7)	36 (46.8)	39 (41.1)	
Married	10 (55.6)	23 (29.9)	33 (34.7)	
Cohabiting	3 (16.7)	12 (15.6)	15 (15.8)	
Divorced	2 (11.1)	6 (7.8)	8 (8.4)	
Hospitalization				2.062 (0.161)
Yes	2 (11.1)	2 (2.6)	4 (4.2)	
No	16 (88.9)	75 (97.4)	91 (95.8)	
Vaccine status				4.990 (0.082)
Unvaccinated	2 (11.1)	21 (27.6)	23 (24.2)	
Incomplete	12 (66.7)	29 (38.2)	41 (43.2)	
Fully vaccinated	4 (22.2)	26 (34.2)	30 (31.6)	
Know what to do in case of suffering discrimination				0.081 (0.781)
No	6 (33.3)	23 (29.9)	29 (30.5)	
Yes	12 (66.7)	54 (70.1)	66 (69.5)	

F. Fisher’s test.

**Table 2 ijerph-22-01775-t002:** Comparison between social exclusion and time with symptoms, measures of care, Pittsburgh quality of sleep, stigma, self-compassion, Type D personality and demoralization.

	Social Exclusion		Total
	Yes = 18 Mean (SD)	No = 77 Mean (SD)	z (*p*)
Weeks with symptoms	3.41 (1.97)	2.32 (1.73)	−3.066 (0.002)
One *	2 (11.1)	32 (41.6)	
More than a week *	16 (88.9)	45 (58.4)	
Care measures	4.34 (0.57)	4.04 (0.78)	−1.528 (0.488)
Yes *	15 (83.3)	59 (76.6)	
No *	3 (16.7)	18 (23.4)	
Sleep quality	6.68 (2.46)	4.87 (2.91)	−2.361 (0.018)
Yes *	6 (33.3)	45 (58.4)	
No *	12 (66.7)	32 (41.6)	
Stigma	14.50 (14.14)	6.51 (9.18)	−2.699 (0.007)
Yes *	6 (33.3)	6 (7.8)	
No *	12 (66.7)	71 (92.2)	
Enacted	5.06 (5.76)	2.79 (4.26)	−2.288 (0.022)
Perceived	9.44 (10.12)	3.72 (5.77)	−2.420 (0.016)
Personality (PTD)	66.50 (12.24)	67.79 (13.78)	−0.542 (0.588)
Yes *	13 (72.2)	61 (79.2)	
No *	5 (27.8)	16 (20.8)	
Positive	33.17 (10.92)	30.82 (9.40)	−0.747 (0.455)
Negative	44.67 (7.87)	41.03 (9.83)	−1.314 (0.189)
Social inhibition	7.39 (4.53)	8.83 (5.91)	−0.800 (0.424)
Negative affect	6.17 (5.24)	7.21 (6.52)	−0.396 (0.692)
Self-Compassion			
Positive	33.17 (10.92)	30.82 (9.40)	−0.925 (0.357)
Negative	44.67 (6.87)	41.03 (9.83)	−1.486 (0.141)
Demoralization	16.33 (11.77)	21.78 (17.33)	−0.926 (0.354)
Yes *	2 (11.1)	15 (19.5)	
No *	16 (88.9)	62 (80.5)	
Helplessness	2.83 (1.89)	4.40 (3.18)	−1.798 (0.072)
Loss of meaning	4.11 (3.38)	5.18 (4.52)	−0.774 (0.439)
Dysphoria	2.27 (2.94)	1.61 (1.50)	−0.010 (0.992)
Sense of failure	5.21 (4.10)	4.22 (4.69)	−1.213 (0.225)
Discouragement	3.56 (4.53)	4.71 (6.47)	−0.024 (0.981)

z. W of Wilcoxon. Enacted. Stigma promulgated Internal. Perceived. Perceived external (social) stigma. Stigma. Stigma total score. Sleep quality. Pittsburgh Sleep Quality Index (ICSP). Personality (PTD). * *n* (%).

**Table 3 ijerph-22-01775-t003:** Demographic and sleep quality variables compared by stigma in patients with COVID-19.

	Stigma Internally Perceived		Stigma Externally Perceived		Total Stigma Score	
	Mean (SD)	z/H (*p*)	Mean (SD)	z/H (*p*)	Mean (SD)	z/H (*p*)
Gender		−0.765 (0.444)		−1.159 (0.246)		1.354 (0.176)
Male	3.05 (4.77)		4.40 (7.63)		7.46 (11.84)	
Female	3.33 (4.47)		5.07 (6.81)		8.40 (9.97)	
Age (years)		0.489 (0.783)		0.575 (0.750)		0.574 (0.750)
18–29	1.91 (2.53)		3.09 (3.73)		5.00 (5.86)	
30–50	3.71 (5.14)		5.49 (8.01)		9.20 (11.93)	
51+	3.08 (4.05)		4.39 (6.64)		7.46 (10.06)	
Education		4.087 (0.130)		2.996 (0.224)		3.456 (0.178)
High school	1.65 (2.81)		3.62 (7.14)		5.27 (8.31)	
University	3.92 (5.15)		5.16 (6. 65)		9.08 (10.98)	
Postgraduate	3.53 (4.56)		5.53 (8.34)		9.05 (12.53)	
Marital status		1.092 (0.779)		2.339 (0.505)		2.506 (0.474)
Single	3.59 (5.74)		5.29 (5.00)		8.16 (5.37)	
Married	3.30 (4.25)		3.28 (4.05)		5.03 (4.36)	
Cohabiting	2.40 (2.03)		3.70 (4.33)		6.65 (5.41)	
Divorced	2.62 (2.77)		3.10 (4.97)		4.57 (5.57)	
Hospitalization		−1.588 (0.112)		−1.677 (0.094)		−1.992 (0.055)
Yes	5.75 (6.18)		15.25 (14.45)		21.00 (17.64)	
No	3.11 (4.50)		4.35 (6.38)		7.46 (10.05)	
Vaccine status		4.715 (0.095)		9.320 (0.009)		7.862 (0.020)
Unvaccinated	1.91 (3.46)		3.30 (7.06)		5.22 (9.44)	
Incomplete	3.97 (4.75)		6.15 (7.61)		10.12 (10.95)	
Fully vaccinated	3.30 (4.98)		4.30 (6.38)		7.60 (11.05)	
Know what to do in case of suffering discrimination		−0.201 (0.840)		−0.172 (0.863)		−0.345 (0.730)
No	2.86 (4.10)		4.62 (7.53)		7.48 (11.40)	
Yes	3.38 (4.78)		4.89 (6.97)		8.28 (10.43)	
ICSP		−3.860 (0.001)		−3.793 (0.001)		−4.254 (0.001)
Good sleep quality	1.68 (2.92)		2.24 (4.08)		3.92 (6.40)	
Poor sleep quality	5.00 (5.45)		7.80 (8.64)		12.80 (12.59)	
Care measures		−0.149 (0.882)		−0.145 (0.885)		−0.114 (0.909)
Yes	3.19 (4.58)		4.93 (7.33)		8.12 (10.82)	
No	3.33 (4.62)		4.38 (6.39)		7.71 (10.43)	

z. W of Wilcoxon. H. Chi-Square. ICSP. Pittsburgh Sleep Quality Index.

**Table 4 ijerph-22-01775-t004:** Model for social rejection in patients with COVID-19.

	*n*	B	ES	Exp (B) 95%	C.I. Para EXP (B)
Variable		−3.648	0.838	0.026	
Gender					
Male	37	1			
Female	58	1.535	0.752	4.640	(1.062–20.266)
Stigma					
No	83	1			
Yes	12	2.205	0.763	9.071	(2.033–40.467)
Vaccination at least with one dose					
No	54	1			
Yes	51	1.304	0.618	3.658	(1.097–12.384)

## Data Availability

The raw data supporting the conclusions of this article will be made available by the authors on request.

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
