# Peer review of "Risk Factors for Social Exclusion and Stigma in a Group of Non-Hospitalised Patients with COVID-19"

_ijerph, 2025, doi:10.3390/ijerph22121775_

Round 1
Reviewer 1 Report (New Reviewer)
Comments and Suggestions for Authors
- The study's title, "Social Exclusion and Stigma in Patients with COVID-19," was too broad and failed to clearly convey the study's subjects, core variables, and their relationship. It lacked the necessary academic focus and distinctiveness. Although the manuscript aimed to investigate the influence of specific risk factors on the social exclusion and stigma experienced by COVID-19 patients—which implies a clear operational and analytical logic—this was not reflected in the title. The authors should have refocused the title to clearly present the study population (e.g., recovered, hospitalized, or home-quarantined individuals), the main variables (e.g., risk factors, social exclusion, stigma), and the investigated relationship (e.g., association, prediction) to enhance readability and academic impact.
- Although the definitions of social exclusion and Stigma were detailed, the introduction lacked empirical literature support concerning social exclusion and Stigma phenomena specifically during the COVID-19 pandemic. Furthermore, the theoretical context and research gaps were not clearly articulated. The research motivation was weak, and the explanation regarding the measures used and their interrelationship was also absent. This resulted in a disjointed and unpersuasive research background. The authors should have strengthened the literature review and theoretical foundation, clearly defining the study's necessity and point of entry.
- The stated goal at the end of the introduction, which was to " describe the risk factors for suffering social exclusion and stigma in a group of people with COVID-19," was clear and academically valuable. However, this goal lacked logical coherence with the generic phrasing of the article title and the result-oriented statements in the abstract. To improve the overall focus and consistency of the manuscript structure, the authors should have guided the reader toward the core research objective of "risk factor analysis" early in the introduction and revised the title and abstract to more precisely reflect the study's central purpose and actual aims.
- The description of the sample source and sampling method was clearly inadequate. Although the authors stated the sample consisted of 95 voluntarily participating COVID-19 patients surveyed via an online questionnaire, they did not specifically detail the recruitment method (e.g., convenience, snowball, or random sampling) or whether clear inclusion and exclusion criteria were established. This omission impacted the representativeness of the sample and the external validity of the results. The authors should have supplemented the manuscript with a detailed description of the recruitment process and sampling strategy to enhance methodological transparency and scientific rigor.
- The actual content of Section "2.2 Sample" should have been titled "Measures," indicating an inappropriate classification and insufficient content description. For the core scales (e.g., Social Exclusion and Stigma Scales), the authors did not clearly explain their original source, subscale structure, number of items, scoring method, or reliability and validity indices (such as Cronbach's α). The background development and validation procedure of the scales were also omitted. It was thus impossible to evaluate their measurement quality and academic validity, particularly since social exclusion was the study's central dependent variable. The authors should have clarified the role of each scale in the study and fully supplemented the source and statistical quality of the measurement tools to enhance the study's rigor and reproducibility.
- The manuscript mentioned using a stigma scale in the materials and methods section, yet the results section focused on a grouped analysis of the "existence of social exclusion." This made it difficult for the reader to determine if these were derived from the same measurement or conceptual indicator, revealing an inconsistency in variable definition and analytical logic. Furthermore, the operational criteria for "determining the existence of social exclusion" (e.g., scale score cutoff point, basis for grouping) were not provided, which lacked a transparent and replicable determination basis. The authors should have clearly explained whether the stigma scale encompassed the social exclusion construct, whether a separate scale was used for social exclusion, and what the determination criteria were, avoiding the blurring of core variable boundaries and enhancing the consistency and academic rigor of the methodology and results interpretation.
- The main flaw in the discussion section was the lack of clear guidance from the research objectives, which resulted in a loose, unsystematic presentation of content that merely listed results and their consistency with the literature. The authors should have systematically restructured this chapter, ensuring each paragraph focused on a core finding (e.g., the influence of gender or vaccination status) and replaced mere description with interpretive and critical depth. The central focus of the discussion should have been on elaborating the theoretical and practical significance of the findings (i.e., explaining "why" these associations were observed), rather than simply demonstrating congruence with previous studies.
- The discussion section failed to systematically match and deeply explain the key risk factors derived from the logistic regression analysis (e.g., gender, education level, source of information). This left the link between the findings and the literature weak and the depth of argumentation insufficient. The authors should have elaborated on the potential mechanisms linking each predictor variable to social exclusion or Stigma, clearly citing empirical or theoretical support, and comparing their results with existing research to note consistency or deviation, thereby strengthening the academic significance and interpretability of the findings.
- The data statements in the discussion section were contradictory to the results of the logistic regression analysis in Table 4. Specifically, the narrative regarding "gender" as a risk factor for social exclusion was the direct inverse of Table 4's results: Table 4 showed that the Odds Ratio (OR) for females was greater than 1 (indicating a risk factor), yet the discussion claimed that males were at a "higher risk."
- The authors stated: "Even though we did not find a direct relation with altered emotional status and stigma it might possible be related." If the statistical results showed no significant association, the authors should not have subjectively speculated or implied a possible relationship in the absence of evidence. The discussion must be strictly based on the actual data obtained. For non-significant results, the authors should have candidly stated that "no evidence supporting an association was found" and attributed this to study limitations (e.g., insufficient sample size or measurement constraints), rather than engaging in unsubstantiated inference.
- The primary deficiency of the discussion section was its failure to elevate the findings to a level of theoretical explanation and practical inspiration, resulting in a lack of the critical analytical depth required by a journal. The discussion merely stated the consistency of results with the literature without fully explaining "why" specific associations were observed (e.g., the mechanisms behind the uncompleted vaccination status or gender differences). Crucially, the authors offered almost no specific inspirations or practical recommendations based on their findings, such as how this knowledge should be applied to public health policies, clinical interventions, or community education to combat Stigma and social exclusion. The authors should have bolstered the discourse on the theoretical significance and practical value of the results to strengthen the study's completeness and societal contribution.
Author Response
Comment 1: The study's title, "Social Exclusion and Stigma in Patients with COVID-19," was too broad and failed to clearly convey the study's subjects, core variables, and their relationship. It lacked the necessary academic focus and distinctiveness. Although the manuscript aimed to investigate the influence of specific risk factors on the social exclusion and stigma experienced by COVID-19 patients—which implies a clear operational and analytical logic—this was not reflected in the title. The authors should have refocused the title to clearly present the study population (e.g., recovered, hospitalized, or home-quarantined individuals), the main variables (e.g., risk factors, social exclusion, stigma), and the investigated relationship (e.g., association, prediction) to enhance readability and academic impact.
Response 1: Thank you for your extensive and justified comments. The title of the article has been modified in accordance with your suggestions and now reads: Risk factors for social exclusion and stigma in a group of non-hospitalised COVID-19 patients.
Comment 2: Although the definitions of social exclusion and Stigma were detailed, the introduction lacked empirical literature support concerning social exclusion and Stigma phenomena specifically during the COVID-19 pandemic. Furthermore, the theoretical context and research gaps were not clearly articulated. The research motivation was weak, and the explanation regarding the measures used and their interrelationship was also absent. This resulted in a disjointed and unpersuasive research background. The authors should have strengthened the literature review and theoretical foundation, clearly defining the study's necessity and point of entry.
Response 2: COVID-19 has had adverse effects on society and individuals. Among other things, stigma and its consequences for mental health have affected people's quality of life. As a social construct, stigma is considered a daily experience of discrimination and rejection [2]. Furthermore, the impact can be long-lasting and continue even when symptoms no longer exist, or the person is no longer contagious to others. For this reason, it is said to be a personal experience or social process characterized by exclusion, rejection, guilt, and devaluation because of negative social judgments.
Comment 3: The stated goal at the end of the introduction, which was to " describe the risk factors for suffering social exclusion and stigma in a group of people with COVID-19," was clear and academically valuable. However, this goal lacked logical coherence with the generic phrasing of the article title and the result-oriented statements in the abstract. To improve the overall focus and consistency of the manuscript structure, the authors should have guided the reader toward the core research objective of "risk factor analysis" early in the introduction and revised the title and abstract to more precisely reflect the study's central purpose and actual aims.
Response 3: Studies on previous outbreaks (SARS, Ebola) and COVID-19 show that a high proportion of survivors report stigmatisation and the perception that they are still contagious, leading their family, friends and neighbours to avoid them (Ho et al., 2020; Ramos et al., 2023). The perception of threat to the population causes segregation and blame during COVID-19, which consisted of rejection of coexistence, denial of services, or exclusion from work. This was directed at healthcare workers, people living on the streets or homeless, migrants, or people of Asian descent. According to the theory of terror management (the psychological need to assign a cause or enemy to the virus), it was based on fear of the unknown and contagion, seeking scapegoats to blame.
Comment 4: The description of the sample source and sampling method was clearly inadequate. Although the authors stated the sample consisted of 95 voluntarily participating COVID-19 patients surveyed via an online questionnaire, they did not specifically detail the recruitment method (e.g., convenience, snowball, or random sampling) or whether clear inclusion and exclusion criteria were established. This omission impacted the representativeness of the sample and the external validity of the results. The authors should have supplemented the manuscript with a detailed description of the recruitment process and sampling strategy to enhance methodological transparency and scientific rigor.
Response 4: Between August and December 2021, a total of 95 convenience surveys were made in patients who had the infection but did not require hospitalisation. Patients over 18 years of age were included, discharged from hospital with a diagnosis of COVID-19 confirmed by PCR test (oropharyngeal swab), who were admitted for moderate SARS-CoV-2 infection according to WHO criteria (moderate disease: presence of clinical signs of pneumonia [fever, cough, dyspnoea and/or tachycardia], without signs of pneumonia [fever, cough, dyspnoea and/or tachycardia], without signs of pneumonia [fever, cough, dyspnoea and/or tachycardia], without signs of pneumonia [fever, cough, dyspnoea and/or tachycardia], without signs of pneumonia [fever, -CoV-2 infection according to WHO criteria (moderate disease: presence of clinical signs of pneumonia [fever, cough, dyspnoea and/or tachycardia], without signs of severe pneumonia and, in particular, oxygen saturation in ambient air [SaO2] ≥ 90%).
Comment 5: The actual content of Section "2.2 Sample" should have been titled "Measures," indicating an inappropriate classification and insufficient content description. For the core scales (e.g., Social Exclusion and Stigma Scales), the authors did not clearly explain their original source, subscale structure, number of items, scoring method, or reliability and validity indices (such as Cronbach's α). The background development and validation procedure of the scales were also omitted. It was thus impossible to evaluate their measurement quality and academic validity, particularly since social exclusion was the study's central dependent variable. The authors should have clarified the role of each scale in the study and fully supplemented the source and statistical quality of the measurement tools to enhance the study's rigor and reproducibility.
Response 5: Section 2.2 Sample was included with the following information: Between June and December 2021, a total of 95 convenience surveys were made in patients who had the infection but did not require hospitalisation. Patients over 18 years of age were included, discharged from hospital with a diagnosis of COVID-19 confirmed by PCR test (oropharyngeal swab), who were admitted for moderate SARS-CoV-2 infection according to WHO criteria (moderate disease: presence of clinical signs of pneumonia [fever, cough, dyspnoea and/or tachycardia], without signs of pneumonia [fever, cough, dyspnoea and/or tachycardia], without signs of pneumonia [fever, cough, dyspnoea and/or tachycardia], without signs of pneumonia [fever, cough, dyspnoea and/or tachycardia], without signs of pneumonia [fever, -CoV-2 infection according to WHO criteria (moderate disease: presence of clinical signs of pneumonia [fever, cough, dyspnoea and/or tachycardia], without signs of severe pneumonia and, in particular, oxygen saturation in ambient air [SaO2] ≥ 90%).
Section 2.3 Battery of Test was also added, with detailed information on the battery applied.
Social exclusion was measured using a subjective relational approach, focusing on the breakdown of social ties (isolation, loneliness), discrimination, and rejection perceived by the individual. In this study, the following questions were asked: "Did you experience any discrimination while you had COVID-19?" and "What happened? What was the situation? Can you describe it?" Those who answered positively and met the subjective relational approach criteria were categorized as experiencing social exclusion. Perceived stigma was assessed using an adapted stigma scale [5, 6], which measures two dimensions: internal stigma (distancing, exclusion, and shame caused by having the disease) and external stigma (degree of discomfort generated in social interactions). It consists of 20 items, each scored on a 4-point Likert scale. Higher scores indicate greater stigmatization. The scale has adequate reliability (Cronbach's alpha = 0.96) and explains 57.5% of the variance. The Self-Compassion Scale (SCS) (26 items) was applied, assessing how the individual perceives their actions in difficult times. It is composed of six domains, in three distinct but theoretically related concepts: common humanity, mindfulness, and self-compassion, which have their counterpart in self-judgment, isolation, and over-identification [14]. An overall SCS rating has adequate reliability and validity in Mexican population internal consistency (Cronbach alpha=0.84) [15,16], even in different cultures [17]. The DS Demoralization Scale (24 items) was also used to explore the inability or incompetence to effectively cope with a stressful situation, through five domains: loss of meaning in life, dysphoria, discouragement, helplessness, and sense of failure [18]. Most studies [19] have found adequate levels of internal consistency (Cronbach alpha=0.94). Type D personality has been described as the trend to experience an increased occurrence of negative affectivity, social inhibition and may negatively affect health status [20]. Poor sleep may cause disadvantageous decision-making and poor social functioning. Pittsburgh Sleep Quality Index (ICSP) assesses the quality of sleep through a self-applicable questionnaire of 24 questions, 19 of which are used to obtain an overall rating, which are rated on a scale of 0 to 3. A score >5 distinguishes those who sleep poorly from those who sleep well (Cronbach alpha = 0.78). [21]
The stigma scale was validated using FACTOR 11.5 software [7], and its component structure was explored using exploratory factor analysis (EFA). Since it is a Likert-type questionnaire, the ordinary least squares method was chosen [7]. Average partial least squares (ALS) was used to determine the number of dimensions; for factor estimation, the unweighted least squares method was used.
Comment 6: The manuscript mentioned using a stigma scale in the materials and methods section, yet the results section focused on a grouped analysis of the "existence of social exclusion." This made it difficult for the reader to determine if these were derived from the same measurement or conceptual indicator, revealing an inconsistency in variable definition and analytical logic. Furthermore, the operational criteria for "determining the existence of social exclusion" (e.g., scale score cutoff point, basis for grouping) were not provided, which lacked a transparent and replicable determination basis. The authors should have clearly explained whether the stigma scale encompassed the social exclusion construct, whether a separate scale was used for social exclusion, and what the determination criteria were, avoiding the blurring of core variable boundaries and enhancing the consistency and academic rigor of the methodology and results interpretation.
Response 6: Social exclusion was measured using a subjective relational approach, focusing on the breakdown of social ties (isolation, loneliness), discrimination, and rejection perceived by the individual. In this study, the following questions were asked: "Did you experience any discrimination while you had COVID-19?" and "What happened? What was the situation? Can you describe it?" Those who answered positively and met the subjective relational approach criteria were categorized as experiencing social exclusion.
Comment 7: The main flaw in the discussion section was the lack of clear guidance from the research objectives, which resulted in a loose, unsystematic presentation of content that merely listed results and their consistency with the literature. The authors should have systematically restructured this chapter, ensuring each paragraph focused on a core finding (e.g., the influence of gender or vaccination status) and replaced mere description with interpretive and critical depth. The central focus of the discussion should have been on elaborating the theoretical and practical significance of the findings (i.e., explaining "why" these associations were observed), rather than simply demonstrating congruence with previous studies.
Response 7: The discussion was restructured to focus on the key findings.
The study aimed to identify risk factors for social exclusion and stigma among non-hospitalized COVID-19 patients, revealing significant psychosocial consequences for those affected. Key findings highlighted the influence of gender, vaccination status, and stigma on social exclusion, providing crucial insights into the interplay between societal perceptions and individual experiences during the pandemic.
Comment 8: The discussion section failed to systematically match and deeply explain the key risk factors derived from the logistic regression analysis (e.g., gender, education level, source of information). This left the link between the findings and the literature weak and the depth of argumentation insufficient. The authors should have elaborated on the potential mechanisms linking each predictor variable to social exclusion or Stigma, clearly citing empirical or theoretical support, and comparing their results with existing research to note consistency or deviation, thereby strengthening the academic significance and interpretability of the findings.
Answer 8: The study identified three key predictors of social exclusion and stigma among non-hospitalized COVID-19 patients.
First, gender emerged as a significant predictor, disproportionately affecting women through social exclusion. This finding aligns with previous research, such as the 2021 EDIS report, which documented increased discrimination against women during the pandemic. [3] This is consistent with the Stigma Linkage Model, which posits that stigma attaches to pre-existing social biases, such as gender inequality, amplifying its impact on marginalized groups.
Comment 9: The data statements in the discussion section were contradictory to the results of the logistic regression analysis in Table 4. Specifically, the narrative regarding "gender" as a risk factor for social exclusion was the direct inverse of Table 4's results: Table 4 showed that the Odds Ratio (OR) for females was greater than 1 (indicating a risk factor), yet the discussion claimed that males were at a "higher risk."
Response 9: The discussion was reviewed, and the inconsistency was removed.
Comment 10: The authors stated: "Even though we did not find a direct relation with altered emotional status and stigma it might possible be related." If the statistical results showed no significant association, the authors should not have subjectively speculated or implied a possible relationship in the absence of evidence. The discussion must be strictly based on the actual data obtained. For non-significant results, the authors should have candidly stated that "no evidence supporting an association was found" and attributed this to study limitations (e.g., insufficient sample size or measurement constraints), rather than engaging in unsubstantiated inference.
Response 10: The claim was removed to avoid making unsubstantiated inferences.
Comment 11: The primary deficiency of the discussion section was its failure to elevate the findings to a level of theoretical explanation and practical inspiration, resulting in a lack of the critical analytical depth required by a journal. The discussion merely stated the consistency of results with the literature without fully explaining "why" specific associations were observed (e.g., the mechanisms behind the uncompleted vaccination status or gender differences). Crucially, the authors offered almost no specific inspirations or practical recommendations based on their findings, such as how this knowledge should be applied to public health policies, clinical interventions, or community education to combat Stigma and social exclusion. The authors should have bolstered the discourse on the theoretical significance and practical value of the results to strengthen the study's completeness and societal contribution.
Response 11: This study provides information on how to address gender disparities, poverty, and stigma to develop public health campaigns. Clinical interventions should aim to raise awareness among both healthcare workers and patients about the detection of stigma and social exclusion. Likewise, community education should be used to eliminate myths, misinformation, and prejudices in order to improve the psychosocial well-being of people affected by COVID-19.
The findings of this study provide critical insights that can inform public health policies, clinical interventions, and community education initiatives to combat stigma and social exclusion among individuals affected by COVID-19.
Reviewer 2 Report (New Reviewer)
Comments and Suggestions for Authors
The phenomenon of social stigma in the context of health issues is characterized by the development of a negative perception of a particular illness in relation to a specific person or a group of individuals with shared characteristics. This approach can have detrimental effects on patients, as well as on their caregivers, family members, and the immediate social surroundings. Consequently, this issue remains pertinent, despite the fact that many pressing aspects of this problem have been addressed over time since the onset of the epidemic.
Below is a detailed list of comments and issues that need to be addressed:
- The theoretical review could be strengthened by incorporating theories and concepts related to social exclusion and stigma. It would be beneficial to include authoritative sources on the psychology of illness and pandemics, among other topics. At the moment, the review appears to be lacking in depth.
- You have not outlined the primary causes of the prejudice surrounding COVID-19. To be precise, the extent of the prejudice related to COVID-19 is influenced by three key factors: a) the novelty of the disease, with many aspects still unknown; b) the natural fear people have of the unknown; c) the emerging fear is often attributed to the machinations of outsiders.
- There is also a question regarding the theoretical link between COVID-19 and stigmatization. According to this line of reasoning, all diseases such as the seasonal flu, pneumonia and others could also be viewed as forms of stigmatization. Therefore, the current argument appears to be unconvincing.
- The declared “aim of the study is to describe the risk factors for suffering social exclusion and stigma”. This matter is not further elaborated upon in the article.
- The choice of methods used in the study is not clearly explained. Why were these particular methods chosen? This again indicates a lack of a thorough theoretical review. The study needs to justify the selection of the methods used.
- The conclusions need revising. Make them persuasive and well-reasoned, rather than merely concise. This will ensure that your piece becomes engaging for a broad audience.
As of now, the manuscript appears to be rather underdeveloped.
Author Response
Comment 1: The theoretical review could be strengthened by incorporating theories and concepts related to social exclusion and stigma. It would be beneficial to include authoritative sources on the psychology of illness and pandemics, among other topics. At the moment, the review appears to be lacking in depth.
Response 1: The emotional nature of fear reinforces stigmatizing beliefs. The fear of death (a central component of TMT) demonstrates the prevalence of mortality awareness, linking stigmatizing behavior to actions aimed at avoiding the fear of death, resulting in greater prejudice and rejection (Ramos Vera). Another study shows how the threat of SARS-CoV-2 quickly translates into xenophobia and prejudice directed toward an out-group perceived as the source (Rzymski and Nowicki).
Comment 2: You have not outlined the primary causes of the prejudice surrounding COVID-19. To be precise, the extent of the prejudice related to COVID-19 is influenced by three key factors: a) the novelty of the disease, with many aspects still unknown; b) the natural fear people have of the unknown; c) the emerging fear is often attributed to the machinations of outsiders.
Response 2: Some research found that fear caused by novelty and emergency was channeled through existing social prejudices (racism, xenophobia, ageism) and misinformation, resulting in discrimination and stigma (Ramos Vera; Rzymski and Nowicki).
Ramos-Vera, C. Relaciones de red del complejo estigma-discriminación y el miedo a la COVID-19 durante la segunda ola pandémica en adultos peruanos. RevCol Psiq. 2023; 52(1), 5–8. https://doi.org/10.1016/j.rcp.2021.05.010
Rzymski P, Nowicki M. COVID-19-related prejudice toward Asian medical students: A consequence of SARS-CoV-2 fears in Poland. J Infect Public Health. 2020;13(6):873-876. https://doi.org/10.1016/j.jiph.2020.04.013
Comment 3: There is also a question regarding the theoretical link between COVID-19 and stigmatization. According to this line of reasoning, all diseases such as the seasonal flu, pneumonia and others could also be viewed as forms of stigmatization. Therefore, the current argument appears to be unconvincing.
Response 3: Several theories explain the relationship between COVID-19 and stigma. Two focus on the fear of infection: one is related to the Behavioral Immune System (BIS), and the other to Terror Management Theory (TMT). BIS is a psychological mechanism that evolved to detect pathogenic signals in the environment (e.g., cough, fever, "unusual" appearances) and motivates avoidance behaviors as a precaution against infection. It is a psychological defense before the biological immune system can act. The person rejects and displays exaggerated social distancing toward any carrier and sometimes overgeneralizes to people who appear ill or belong to risk groups, even if they are not a real threat. In TMT, the awareness of one's own mortality drives prejudice and the need for group belonging. During COVID-19, the high number of deaths reinforces identification with one's own group and hostility toward what is considered or perceived as a threat, giving rise to xenophobia. In Labeling Theory (LT), stigma arises when a person or group is negatively labeled, leading to social exclusion and internalizing shame. The Stigma Linkage Model explains how stigma is linked to pre-existing social prejudices (race, class, geography, political orientation, etc.), as it does not occur in a vacuum but rather adheres to established social divisions and inequalities. Thus, stigmatization during COVID-19 is a complex response driven by primal fear of infection and death (BIS, TMT), reinforced by social labels (LT), and directed toward groups already vulnerable due to pre-existing prejudices (SLM).
Comment 4: The declared “aim of the study is to describe the risk factors for suffering social exclusion and stigma”. This matter is not further elaborated upon in the article.
Response 4: Studies on previous outbreaks (SARS, Ebola) and COVID-19 show that a high proportion of survivors report stigmatisation and the perception that they are still contagious, leading to avoidance by family, friends and neighbours (Ho et al., 2020; Ramos et al., 2023).The perception of threat to the population led to segregation and blame during the second wave of COVID-19, which consisted of rejection of coexistence, denial of services, or exclusion from work. This was directed at healthcare workers, people living on the streets or who were homeless, migrants, or people of Asian descent. According to the theory of terror management (the psychological need to assign a cause or enemy to the virus), it was based on fear of the unknown and contagion, seeking scapegoats to blame.
Comment 5: The choice of methods used in the study is not clearly explained. Why were these particular methods chosen? This again indicates a lack of a thorough theoretical review. The study needs to justify the selection of the methods used.
Response 5: The methods used in the study were reviewed and clarified in the statistical analysis section.
Comment 6: The conclusions need revising. Make them persuasive and well-reasoned, rather than merely concise. This will ensure that your piece becomes engaging for a broad audience.
Response 6: In essence, the stigmatization during this period was based on fear of the unknown and of contagion, which translated into a search for scapegoats and the amplification of pre-existing prejudices. Thus, stigmatization during COVID-19 is a complex response driven by primal fear of infection and death (BIS, TMT), reinforced by social labels (LT), and directed toward groups already vulnerable due to pre-existing prejudices (SLM).
We hope that with these changes, the work will better meet expectations, be more justifiable, and be more engaging.
Round 2
Reviewer 2 Report (New Reviewer)
Comments and Suggestions for Authors
The authors have made significant improvements to the quality of the manuscript. Nevertheless, line 51 in the Introduction section and line 309 of the Conclusions section requires further refinement to ensure a more academic tone. Specifically, the expression “scapegoat search” should be removed.
Author Response
Comments 1: The authors have made significant improvements to the quality of the manuscript. Nevertheless, line 51 in the Introduction section and line 309 of the Conclusions section requires further refinement to ensure a more academic tone. Specifically, the expression “scapegoat search” should be removed.
Response 1: Thank you for your comments. We have made changes to improve the writing style, making it more academic in the sentences you pointed out.
Line 51: According to the theory of terror management (the psychological need to assign a cause or enemy to the virus), it was based on fear of the unknown and contagion, looking for someone to blame.
Line 309: In essence, the stigmatization during this period was based on fear of the unknown and of contagion, which translated into a search for infected people to blame and the amplification of pre-existing prejudices.
This manuscript is a resubmission of an earlier submission. The following is a list of the peer review reports and author responses from that submission.
Round 1
Reviewer 1 Report
Comments and Suggestions for Authors
Dear Authors
Thank you for the opportunity to read and review this work.
Congratulation on your work. However it is necessary some improvements.
The theoretical background presented by the authors it is incipient for the study. I recommend that this section be improved. For example, this theoretical background does not support the discussion about the consequences of the Social Exclusion and Stigma in Patients with COVID-19. It's the important results of this study, so the theoretical background should allow understanding not only its relevance but also future discussion and interpretation. Additional, the actual state of theoretical background do not permit us understand the choice of study variables. By example: why the author measures the quality of sleep or self-compassion? What it is important?
Something similar happens with the objective of the study. The objective of study that the authors present (line 68 – “The objective of this study was to know and describe the frequency of social exclusion in people suffering from COVID-19 in Mexico”) it is reductive for the data that later present. This objective needs to be revised and extended to represent what the authors actually present.
The section “2. Materials and Methods” it is poorly structured and incomplete. This section should have the following subsections: Procedures, Sample, Measures, Statistical analysis and Ethical Consideration - some of them are missing now. The data about sample should be present in this section not in results. The readers should have access to Cronbach alpha of all scales.
The discussion performed by the authors needs to be better and clear. The authors present more conclusions from other studies than reflect your results.
By example (line 150-151: “they lived involuntary isolation as a traumatic experience”)- this sentence is decontextualized, the results of this study do not reflect the traumatic experience associated to involuntary isolation.
Line 158-160 – “Discrimination and social exclusion may have detrimental consequences and reduce the possibility of seeking help because of social stigma”, more one time this not evaluated in this study, this is a bad extrapolation. And “On the other hand, they also had longer symptoms than people who did not report suffering social exclusion” but what is mean? In your opinion social exclusion can improver longer symptoms or people with longer symptom are more vulnerable to social exclusion? And Why? This reflection is important.
Line 165 – 167 - “This also shows effects after the suffering of COVID-19, because 46.3% report poor sleep quality, being one of the most frequent sequelae of the disease.” – the poor sleep quality is result of suffering covid-19 or social exclusion? Why do you measure sleep quality? What did you expect about it?
Women show more social exclusion than man, it is congruent with other studies, but what is mean? Why women are more vulnerable in this case?
These is only some examples in your discussion. You need to discuss more your findings, interpret your findings, not the other results. In this form it isn’t possible to identify your contribution beyond the study of other authors.
The work needs to present theoretical and practical implications and, limitations.
Author Response
Reviewer 1
Dear Authors
Thank you for the opportunity to read and review this work.
Congratulation on your work. However it is necessary some improvements.
The theoretical background presented by the authors it is incipient for the study. I recommend that this section be improved. For example, this theoretical background does not support the discussion about the consequences of the Social Exclusion and Stigma in Patients with COVID-19. It's the important results of this study, so the theoretical background should allow understanding not only its relevance but also future discussion and interpretation. Additional, the actual state of theoretical background do not permit us understand the choice of study variables. By example: why the author measures the quality of sleep or self-compassion? What it is important?
Something similar happens with the objective of the study. The objective of study that the authors present (line 68 – “The objective of this study was to know and describe the frequency of social exclusion in people suffering from COVID-19 in Mexico”) it is reductive for the data that later present. This objective needs to be revised and extended to represent what the authors actually present.
Response: Many thanks for your time and comment. The objective sentence was reformatted to be congruent with the whole paper.
The section “2. Materials and Methods” it is poorly structured and incomplete. This section should have the following subsections: Procedures, Sample, Measures, Statistical analysis and Ethical Consideration - some of them are missing now. The data about sample should be present in this section not in results. The readers should have access to Cronbach alpha of all scales.
Response: Many thanks for your time and comment. The Materials and methods sections were restructured, and the requested information was completed.
The discussion performed by the authors needs to be better and clear. The authors present more conclusions from other studies than reflect your results.
By example (line 150-151: “they lived involuntary isolation as a traumatic experience”)- this sentence is decontextualized, the results of this study do not reflect the traumatic experience associated to involuntary isolation.
Line 158-160 – “Discrimination and social exclusion may have detrimental consequences and reduce the possibility of seeking help because of social stigma”, more one time this not evaluated in this study, this is a bad extrapolation. And “On the other hand, they also had longer symptoms than people who did not report suffering social exclusion” but what is mean? In your opinion social exclusion can improver longer symptoms or people with longer symptom are more vulnerable to social exclusion? And Why? This reflection is important.
Line 165 – 167 - “This also shows effects after the suffering of COVID-19, because 46.3% report poor sleep quality, being one of the most frequent sequelae of the disease.” – the poor sleep quality is result of suffering covid-19 or social exclusion? Why do you measure sleep quality? What did you expect about it?
Women show more social exclusion than man, it is congruent with other studies, but what is mean? Why women are more vulnerable in this case?
Response: Many thanks for your time and comment. The discussion section was improved in order to complete the requested information.
These is only some examples in your discussion. You need to discuss more your findings, interpret your findings, not the other results. In this form it isn’t possible to identify your contribution beyond the study of other authors.
The work needs to present theoretical and practical implications and, limitations.
Response: Many thanks for your time and comment. We made changes to complete all your suggestions.

Reviewer 2 Report
Comments and Suggestions for Authors
The article addresses an important issue: the social exclusion and stigma that people infected with SARS-COV2 may suffer. However, the article fails to make its objective clear. In the Abstract the authors state that: "The aim of the study was to describe the risk factors for suffering social exclusion and stigma in a group of people with COVID-19", but in the Introduction they say "The objective of this study was to know and describe the frequency of social exclusion in people suffering from COVID-19 in Mexico." Clarifying the objective will provide a thread to the article.
Furthermore, the study does not provide a clear definition of social exclusion and how it was measured, which may limit the interpretation of the results. Another key concept that needs explanation is that of stigma, more precisely the distinction the authors make between internally and externally perceived stigma. This distinction is indispensable for the correct interpretation of Table 3.
In the Discussion it is not always easy to distinguish when the authors are referring to the results of their study or other research.
The authors should mention the limits of their study, namely, the size of their sample.
Author Response
Reviewer 2
The article addresses an important issue: the social exclusion and stigma that people infected with SARS-COV2 may suffer. However, the article fails to make its objective clear. In the Abstract the authors state that: "The aim of the study was to describe the risk factors for suffering social exclusion and stigma in a group of people with COVID-19", but in the Introduction they say "The objective of this study was to know and describe the frequency of social exclusion in people suffering from COVID-19 in Mexico." Clarifying the objective will provide a thread to the article.
Response: Many thanks for your time and comment. The objective of this study was to describe the risk factors for suffering social exclusion and stigma in a group of people with COVID-19.
Furthermore, the study does not provide a clear definition of social exclusion and how it was measured, which may limit the interpretation of the results.
Another key concept that needs explanation is that of stigma, more precisely the distinction the authors make between internally and externally perceived stigma. This distinction is indispensable for the correct interpretation of Table 3.
Response: Many thanks for your time and comment. Internal stigma (refers to the distance, exclusion and shame caused by suffering from the disease), and externally perceived stigma (degree of discomfort generated in social interactions). Cronbach alpha=0.96. We explained in more detail the requested definitions.
In the Discussion it is not always easy to distinguish when the authors are referring to the results of their study or other research.
Response: Many thanks for your time and comment. The discussion section was improved in order to complete the requested information.
The authors should mention the limits of their study, namely, the size of their sample.
Response: Many thanks for your time and comment. We added our study limitations as you suggested.
Round 2
Reviewer 2 Report
Comments and Suggestions for Authors
The authors have made important corrections in this version of the paper, but the text still needs to be revised by a native speaker.